# Knowledge Guided Bayesian Flow Network for CAD Sequence Generation

## Abstract

The controllable generation of parametric CAD sequences under explicit quantitative constraints (e.g., surface area, volume) is crucial for automating design processes, as it enables the efficient and precise creation of complex geometric models that meet predefined functional or physical requirements. However, this task remains highly challenging due to the multimodal nature of CAD data, which combines discrete commands and continuous parameters, as well as the long-range dependencies among parameters that are critical for satisfying the constraints. While deep generative models have shown remarkable progress in various domains, they still struggle with parametric CAD sequence generation under strict quantitative constraints. To tackle this, we propose a generative framework based on a Knowledge-Guided Bayesian Flow Network (KGBFN). Our approach leverages Bayesian flow to jointly model discrete and continuous variables, effectively capturing the complex structure of CAD data. Moreover, we introduce a knowledge-guided Bayesian update strategy that iteratively injects property constraints during the generation process, significantly enhancing the accuracy of the produced sequences. To improve computational efficiency, we design a dual-channel Bayesian flow network that integrates both traditional and knowledge-guided updates, employing an annealing mechanism to dynamically control the activation of different channels. This design effectively balances knowledge guidance with optimization efficiency. We validate our method on CAD generation tasks constrained by quantitative properties such as surface area and volume. Experimental results demonstrate that our model consistently outperforms state-of-the-art methods in both single and multi-condition constrained generation, achieving superior performance in terms of accuracy and feasibility.

## 1 Introduction

As a core technology in intelligent manufacturing, computer-aided design (CAD) plays a crucial role in automating design processes and accelerating optimization iterations. Currently, generating high-fidelity parametric CAD sequences under explicit quantitative constraints (such as dimensions, volume, surface area, etc.) has become a central challenge in advancing intelligent design. This challenge primarily stems from the complex interplay between discrete operation commands and continuous design parameters in CAD data, as well as the stringent requirements that constraints impose on the precision and feasibility of sequence generation.

Existing methods for CAD sequence generation have evolved through three stages: (1) Early rule-based systems encoded expert knowledge but lacked scalability Hoshi et al. (2000); (2) Deep geometric generation methods automated modeling but often neglected design-for-manufacturing (DFM) principles Wu et al. (2016); Jiménez et al. (2019); (3) NLP-CAD approaches treated CAD models as parametric sequences, yet struggled with ambiguities in instruction-geometry mapping Wu et al. (2021); Willis et al. (2021b); Dupont et al. (2024).

Recent studies have focused on leveraging advanced generative models for CAD sequence generation, with Transformer-based autoregressive models and diffusion models (DMs) emerging as two promising approaches. While Transformer-based autoregressive models Yang et al. (2019); Vaswani et al. (2017) have shown promise in capturing long-range dependencies, they suffer from error accumulation due to sequential prediction. On the other hand, diffusion models (DMs) Ho et al. (2020);

Dhariwal & Nichol (2021) improve generation quality through iterative denoising, but encounter two fundamental challenges when applied to CAD generation. First, the discrete nature of CAD commands conflicts with the continuous noise assumption inherent in diffusion processes Alam & Ahmed (2024). Second, noise injection can amplify parameter sensitivity Wilson et al. (2024), making it difficult to maintain critical dimensional tolerances. Some approaches have attempted to address these issues, such as CAD-Diffuser Ma et al. (2024), which uses a dual-branch architecture to separate discrete and continuous data, and CADiffusion Bai et al. (2024), which incorporates a geometry-aware decoder to guide noise prediction. However, CAD-Diffuser's complex structure may introduce artifacts Ma et al. (2024), while CADiffusion's constraint handling is limited to simple geometric relationships, rendering it insufficient for complex assembly constraints.

In this paper, we propose a generative framework via a Knowledge-Guided Bayesian Flow Network (KGBFN) for high-fidelity generation of parametrized CAD sequences under quantitative constraints. The proposed framework integrates real-time constraint feedback into the generation process. Leveraging Bayesian flow, it models discrete and continuous variables to capture the complex structure of CAD data. We introduce a knowledge-guided Bayesian update strategy that injects property constraint knowledge during iterative generation, enhancing sequence accuracy. To further optimize efficiency, we design a dual-channel Bayesian flow network that combines traditional Bayesian updates with knowledge-guided updates, employing an annealing mechanism to control the activation of different channels dynamically. This dual-channel approach balances knowledge guidance and optimization efficiency. We validate our approach on a CAD dataset specifically constructed with explicit quantitative constraints, focusing on surface area and volume. Experimental results demonstrate that KGBFN consistently outperforms state-of-the-art methods in both single-condition and multi-condition constrained generation tasks, achieving superior accuracy and robustness. In summary, our main contributions are as follows:

- We propose a novel generative framework using a Knowledge-Guided Bayesian Flow Network (KGBFN) that integrates geometric constraint knowledge via a knowledge-guided Bayesian update, enabling high-fidelity parametric CAD sequence generation under quantitative constraints.

- We design a dual-channel Bayesian flow network that balances Bayesian updates with knowledge-guided updates, utilizing an annealing mechanism to dynamically control channel activation, thereby improving computational efficiency.

- Comprehensive experimental results on quantitatively constrained CAD generation tasks demonstrate the superior effectiveness of our approach compared to existing methods.

## 2 RELATED WORK

### 2.1 PARAMETRIC CAD MODELING

In the field of CAD generation, current research methodologies can be primarily categorized into the following approaches: parametric CAD generation Wu et al. (2021); Xu et al. (2022); Willis et al. (2021a), topology-free point cloud generation Rao et al. (2020); Tang et al. (2022); Uy et al. (2022), voxel-based 3D representation Uy et al. (2022); Peng et al. (2025), boundary representation (B-Rep) generation Colligan et al. (2022); Jayaraman et al. (2021; 2022), text-guided CAD Generation You et al. (2024); Khan et al. (2024); Li et al. (2024), and 2D sketch generation Xu et al. (2021); Seff et al. (2021). Parametric CAD modeling offers a compact, interpretable, and editable representation of 3D designs, where geometric structures are described as sequences of operations with discrete commands and continuous parameters. This formulation not only enables efficient storage and editing compared to low-level representations such as voxels or point clouds, but also directly aligns with industrial CAD workflows, making it particularly valuable for downstream tasks such as design automation, optimization, and manufacturability analysis. In recent years, with the establishment of large-scale parametric CAD datasets Koch et al. (2019); Willis et al. (2021b), significant progress has been made in parametric CAD sequence modeling Jayaraman et al. (2022); Guo et al. (2022); Wang et al. (2022).

Current research primarily follows two technical directions: unconditional generation methods focus on producing high-quality and diverse 3D models Sharma et al. (2020); Smirnov et al. (2019); Wang et al. (2020), while conditional generation methods are predominantly oriented toward 3D

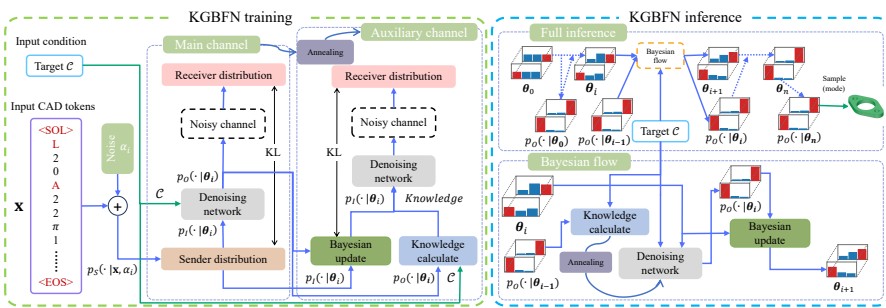

Figure 1: Method overview.The model adopts a dual-channel Bayesian flow structure composed of a main channel and an auxiliary channel, with an annealing mechanism used to dynamically select the active channel during training and inference. The main channel follows the standard BFN training and inference process, while the auxiliary channel incorporates knowledge guidance, which significantly enhances the model's performance in constraint alignment and generation accuracy, albeit at the cost of increased computational overhead.

CAD reconstruction tasks, aiming to convert various geometric inputs into precise CAD operation sequences. These approaches typically encompass point cloud reconstruction (e.g., DeepCAD)Wu et al. (2021), partial CAD model completion (e.g., HNC-CAD)Xu et al. (2023), boundary representation or voxel grid reconstruction (e.g., SECAD)Li et al. (2023), text- or image-guided generationXu et al. (2021); You et al. (2024), and sketch reconstructionRao et al. (2020), among other input forms. It is particularly noteworthy that existing conditional generation methods primarily handle qualitative inputs, while research on quantitatively constrained conditional generation remains largely unexplored. To address this gap, this study contributes to this emerging field by constructing a dedicated dataset and systematically investigating the application of quantitative constraints in parametric CAD sequence modeling.

## 2.2 BAYESIAN FLOW NETWORK

Bayesian Flow Networks (BFNs) Graves et al. (2023) constitute a novel probabilistic generative framework that combines Bayesian inference with continuous flow modeling. Its key advantage lies in directly modeling the underlying data distribution rather than the data itself, contrasting with diffusion models Ho et al. (2020); Dhariwal & Nichol (2021); Rombach et al. (2022) that operate through iterative denoising processes. By preserving the continuity and differentiability of probability distributions, BFNs effectively address the modeling challenges of discrete data types (e.g., text). Previous research has mathematically established the equivalence between BFNs and diffusion models from the perspective of stochastic differential equations Xue et al. (2024), while GeoBFN Ni et al. (2025) demonstrated that BFNs exhibit less inductive bias than diffusion models, making them particularly suitable for noise-sensitive data. MolCRAFT Qu et al. (2024) identified that traditional BFNs apply excessively strong Gaussian noise during inference and proposed corresponding modifications. Notably, existing denoising models universally rely on initial generation conditions during denoising cycles. To address this, our study enhances denoising capability by optimizing long-term dynamic conditional dependencies within the denoising loop, thereby filling this research gap.

## 3 PRELIMINARY

### 3.1 PROBLEM DEFINITON

Parametric CAD modeling is a computer-aided design paradigm that represents geometry through a sequence of parameterized operations. Each operation combines discrete commands (e.g., extrusion, rotation, Boolean operations) with continuous parameters (e.g., length, radius, angle), allowing a 3D model to be defined not as raw point clouds, meshes, or voxels, but as an editable parametric sequence where every step can be traced and modified. In this paper, we address the problem of generating parametric CAD sequences under quantitative geometric constraints. Formally, we define a parametric CAD sequence as: $\boldsymbol{x} = (x_1, \ldots, x_D) \in \{1, \ldots, K\}^D$, where $K$ is the size of

the operation command vocabulary and $D$ denotes the maximum sequence length. The dataset used in our experiments is based on the DeepCAD dataset. We represent the conditional constraints as a vector $C \in \mathbb{R}^d_+$, where $d$ is the number of constraint dimensions and each element lies in the positive real domain.

The mainstream method DeepCAD treats each parameterized CAD instruction as an individual token and handles variable-length instructions using padding and embedding techniques. In contrast, our approach adopts a simpler strategy that only involves padding and concatenation. Specifically, all parameterized instructions are sequentially concatenated into a continuous discrete token sequence, which is then padded to obtain a fixed-length input, as illustrated in the lower-left part of Figure 1. This representation is more aligned with natural language modeling paradigms, enabling the model to leverage the strengths of sequence modeling in capturing contextual dependencies and complex interactions among instructions, improving generation quality and constraint satisfaction.

Given this formulation, the goal is to learn a conditional generator function $f$ that maps a geometric constraint vector $C$ to a valid parametric CAD sequence $\boldsymbol{x} = f(C)$. The generator must satisfy two key requirements: (1) the generated sequence $\boldsymbol{x}$ must follow the syntactic rules of the CAD system, and (2) the resulting CAD model instantiated from $\boldsymbol{x}$ must satisfies all constraints in $C$.

### 3.2 BAYESIAN FLOW NETWORK

The Bayesian Flow Network (BFN) constructs a probabilistic modeling system through five core components: (1) the input distribution $p_I$ serving as the prior probability state $\theta_0$, (2) the output distribution $p_O$ as the neural network's ($\Phi(\cdot)$) prediction, (3) the sender distribution $p_S$ implementing data noising, (4) the receiver distribution $p_R$ constructing noisy predictions, and (5) the Bayesian update function enabling parameter transfer. This framework demonstrates unique advantages in discrete data modeling.

In discrete data modeling, the Bayesian flow network parameterizes the distributions as $\theta_i \in \mathbb{R}^{D \times K}$, where each row $\theta_i^d \in \Delta^{K-1}$ corresponds to the categorical distribution at the d-th token position. For a discrete symbol sequence $\mathbf{x} \in \{1, \ldots, K\}^D$, the sender distribution is defined as a conditional probability distribution that injects Gaussian noise into the real data:

$$p_S(\mathbf{y}|\mathbf{x}; \alpha) = \prod_{d=1}^{D} p_S(y^{(d)}|x^{(d)}; \alpha), \tag{1}$$

where $\alpha$ controls the intensity of the Gaussian noise.

The output distribution is derived from the input distribution (noisy belief state $\boldsymbol{\theta}$ from sender) via the time-dependent denoising network $\Phi$:

$$p_O(x|\theta, t) = \prod_{d=1}^{D} p_O\left(x^{(d)}|\Phi^{(d)}(\theta, t)\right) \tag{2}$$

Although each dimension $x^{(d)}$ is sampled independently from its corresponding logits $\phi^{(d)}$, the shared network $\Phi$ generates all logits from the full belief state $\boldsymbol{\theta}_i$, thereby establishing implicit cross-dimensional dependencies. During the loss computation phase, Bayesian Flow Networks establish noise-consistent learning through the construction of a receiver distribution. This distribution forms a parametric estimate of the true data distribution by applying Gaussian noise with the same intensity as the sender's to the output distribution, formally defined as:

$$p_R(y|\theta; t, \alpha) = \mathbb{E}_{p_O(x|\theta; t)} [p_S(y|x; \alpha)] \tag{3}$$

The network is trained by minimizing the KL divergence between the sender and receiver distributions, with the loss function defined as Kullback & Leibler (1951):

$$\mathcal{L} = D_{\mathrm{KL}}\Big(p_S(\mathbf{y}|\mathbf{x}) \,\|\, p_R(\mathbf{y}|\boldsymbol{\theta})\Big), \tag{4}$$

where $p_S$ represents the noise-corrupted data distribution constructed by the sender and $p_R$ denotes the denoised predictive distribution from the receiver. This distribution-matching training approach theoretically ensures consistency between the generation and Bayesian inference. Moreover, during inference, the Bayesian update function facilitates parameter transfer across timesteps. By combining samples from the output distribution $p_O$ with the previous input distribution parameters $\theta_{i-1}$, it

generates the input distribution parameters $\theta$ for the next timestep. Formally, this update is:

$$p_U(\boldsymbol{\theta} \mid \boldsymbol{\theta}_{i-1}, \boldsymbol{x}; \alpha) = \mathbb{E}_{\mathcal{N}(y|\alpha(Ke_x-1),\alpha KI)}\left[\delta\left(\boldsymbol{\theta} - \frac{e^y \boldsymbol{\theta}_{i-1}}{\sum_{k=1}^{K} e^{y_k}(\boldsymbol{\theta}_{i-1})_k}\right)\right] \tag{5}$$

where $\boldsymbol{x}$ is observed data sampled from the output distribution $p_O$, $K$ is the vocabulary size.

## 4 METHODS

To effectively address the quantitatively constrained CAD generation problem, we propose a Knowledge-Guided Bayesian Flow Network (KGBFN) tailored to generate the CAD sequences (Figure 1). The KGBFN framework incorporates three advanced strategies for precise control over parametric CAD modeling: knowledge-guided bayesian update function, dual-channel bayesian flow, and annealing routing mechanism. The rest of this section will elaborate on these techniques.

### 4.1 KNOWLEDGE-GUIDED BAYESIAN UPDATE FUNCTION

While classic Bayesian Flow Networks work well for discrete CAD modeling, they struggle to enforce quantitative constraints during long-sequence denoising process, which reduce accuracy of the generated CAD sequences. The standard Bayesian update process typically involves two key steps: (1) computing the output distribution $p_O(x|\theta)$ via a neural network $\Phi$, and (2) updating the input distribution $p_I(x|\theta)$ based on the output distribution through the Bayesian update function. Although effective in theory, this approach lacks a way to incorporate constraints during generation.

To address this, we introduce a knowledge-guided mechanism that integrates quantitative constraint feedback during intermediate generation steps. Specifically, given a valid intermediate result $k_i$ (a parametric CAD sequence) obtained during the denoising step $t_i$, we quantify the discrepancy between the rendered CAD model's geometric properties and the target constraints as knowledge guidance. This enables us to reformulate the output distribution calculation, where both quantitative constraints and knowledge guidance jointly shape the distribution parameters.

Assuming that $V(k_i, C)$ is a metric function that quantifies the geometric deviation between the intermediate result $k_i$ and target constraint $C$, we can formally express this as:

$$V_i(k, C) = f_{\text{geom}}(k_i) - C, \tag{6}$$

Here, $f_{\text{geom}}$ denotes a function that extracts $m$-dimensional geometric features (e.g., surface area and volume) from the parameterized CAD sequence $k_i$. These features are obtained by rendering the CAD model using the PythonOCC Paviot (2022) library, making the function non-differentiable. Then the output distribution calculation (Eq. 2) is extended as follows:

$$p_O(x|\theta_i, t_i, C, V_i) = \prod_{d=1}^{D} \text{softmax}\left(\Phi^{(d)}(\theta_i, t_i, C, V_i)\right) \tag{7}$$

Based on this, the receiver and sender distributions (Eq 3) can be expressed as:

$$p_R(y_i|\theta_i, t_i, \alpha_i, C, V) = \mathbb{E}_{p_O(x'|\theta_i; t_i, C, V)}\left[p_S(y_i|x'; \alpha_i, C)\right] \tag{8}$$

This improvement introduces a constraint function $V(\cdot)$, enabling the receiver distribution to better align with the characteristics of the sender distribution, with the following theoretical guarantee:

**Theorem 1.** *(Proof in Appendix A) Let $C(\cdot)$ be a function that measures the desired property of the data and $V(\cdot)$ be a known function. If the correlation coefficient $Corr(C, V) \geq 0$, we have:*

$$D_{KL}\left(p_S(y|x, \alpha, C) \parallel p_R(y|\theta, \alpha, C, V)\right)$$
$$\leq D_{KL}\left(p_S(y|x, \alpha, C) \parallel p_R(y|\theta, \alpha, C)\right) \tag{9}$$

This inequality indicates that incorporating the constraint function $V(\cdot)$ into the receiver distribution reduces the KL divergence between the sender and receiver distributions, thereby enhancing the alignment between them.

The complete training and sampling procedures are detailed in Appendices C.1–C.2. Compared to the standard BFN (Algorithms B.1–B.2 in the appendix), our method introduces a knowledge

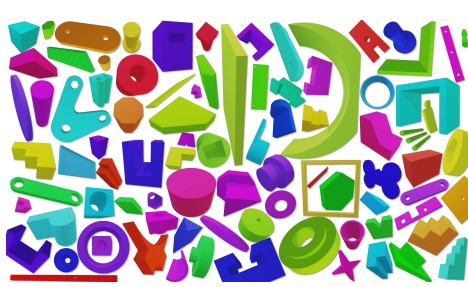

Figure 2: Overview of the parts dataset.

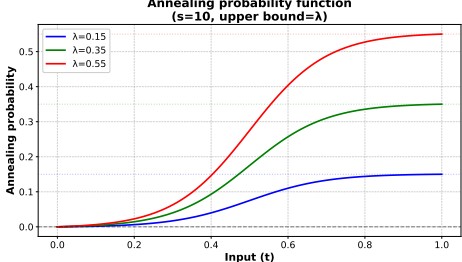

Figure 3: Overview of annealing probability function.

guidance mechanism (line 19) that dynamically adjusts the output distribution parameters based on the preceding denoising step, significantly enhancing the quality of the generated results.

## 4.2 DUAL-CHANNEL BAYESIAN FLOW

In conventional Bayesian Flow Networks, input distributions are typically obtained by either: (1) adding noise to real data or (2) performing Bayesian updates with output distributions. However, both methods often produce input distributions with significant random deviations from real data. In contrast, the output distributions generated by denoising networks demonstrate considerably smaller deviations, especially under KL loss constraints. For parametric CAD sequence generation, our analysis reveals that as model training progresses, using input distributions as intermediate results $k_i$ results in nearly zero validity, while using output distributions yields validity rates as high as 99%. Motivated by this observation, we propose a dual-channel Bayesian flow framework that utilizes output distributions as intermediate results $k_i$ to facilitate knowledge-guided generation.

As shown in Figure 2, our framework uses a dual-channel architecture during training. The main channel follows the original BFN path, producing output distribution $p_O(x|\theta_i, t_i, C)$, input parameters $\theta$, and main KL loss $L_{\text{main}}$. The auxiliary channel takes these outputs, computes knowledge guidance $V$ from $p_O$ and constraints $C$, and transforms them into auxiliary parameters $\theta_{i+1}$ via a Bayesian update. During training, the denoising network $\Phi$ processes $\theta_{i+1}$, $C$, and $V_i$ to compute auxiliary loss $L_{\text{aux}}$. During inference, only one channel is used per step. When the auxiliary channel is active, its intermediate variable $k_i$ is directly taken from the previous step's output distribution.

The main and auxiliary channels share the same denoising network $\Phi$ but differ in their input configurations, which are specified as follows:

$$p_O(x|\theta, t) = \begin{cases} \Phi\big(\theta_i, t_i, C, 0, 0\big) & \text{main,} \\ \Phi\big(\theta_i, t_i, C, V(k, C), 1\big) & \text{aux.} \end{cases} \tag{10}$$

Here, the last two variables represent the knowledge guidance term and the channel indicator, respectively. In the main channel, both are set to 0, indicating the use of a standard Bayesian update without knowledge guidance. In the auxiliary channel, the knowledge guidance term $V(k, C)$ incorporates constraint information, while the channel indicator is set to 1, specifying that the knowledge-guided Bayesian update strategy is applied.

## 4.3 ANNEALING ROUTING MECHANISM

In the auxiliary channel, knowledge computation requires large-scale rendering of parametric CAD sequences into models, creating substantial overhead. To address this, we propose an annealing mechanism that dynamically adjusts the auxiliary channel's activation probability $f(t)$ based on the denoising step. This preserves knowledge guidance benefits while significantly improving training and inference efficiency.

The activation probability is defined as follows:

$$f(t) = \lambda \cdot \frac{\sigma\big(s \cdot (t - 0.5)\big) - \sigma(-0.5 \cdot s)}{\sigma(0.5 \cdot s) - \sigma(-0.5 \cdot s)}. \tag{11}$$

Here $\sigma(\cdot)$ denotes the standard sigmoid function and $t \in [0, 1]$ represents the normalized denoising timestep. The parameter $s$ controls the steepness of the transition region, where a larger value of $s$ results in sharper transitions in the activation probability. The coefficient $\lambda$ serves as an upper bound, ensuring that the output strictly lies within the range $[0, \lambda]$. The function curve is depicted in Figure 3. Based on the calculated activation probabilities, we construct a binary sampler that determines whether to activate the auxiliary channel. The auxiliary channel is triggered when the sampler outputs a 1, and remains inactive otherwise.

During training, after the main channel computes, the system uses the timestep $t$ to decide if the auxiliary channel needs to compute for each sample. During inference, starting from the second denoising step, it dynamically chooses between the main and auxiliary channels for each step.

## 5 EXPERIMENTS

### 5.1 DATASET CONSTRUCTION

To validate the CAD generation method under quantitative constraints, this study constructs a dataset that precisely associates geometric parameters with CAD sequences, where each CAD model is annotated with its key geometric parameters (surface area $\mathcal{A}$ and volume $\mathcal{V}$). Based on 178,238 original parametric CAD models Wu et al. (2021), geometric properties are calculated using the B-Rep analysis module of PythonOCC Paviot (2022), and samples with parametric sequence lengths $\leq$ 64 tokens are selected while duplicate models are removed. Figure 2 presents a subset of the chosen 3D models. Ultimately, a training/validation/test set split of 68,219/6,327/5,652 was obtained, achieving precise mapping between CAD sequences and geometric constraints.

Analysis of 178,238 3D models shows: Both surface area (median = 1.78 $m^2$, skewness = 1.43) and volume (median = $6.81 \times 10^{-2}$ $m^3$, skewness = 3.84) distributions are right-skewness (skewness > 1), indicating a predominance of small-scale models. The normalized dispersion, measured by the interquartile range (IQR) relative to the median, shows that surface area exhibits 91% greater relative variability than volume (2.92 vs. 1.53). Moreover, surface area and volume demonstrate a strong linear relationship, with a Pearson correlation coefficient of $r = 0.82$ ($p \leq 0.001$). These statistical findings confirm the geometric consistency of the dataset and the reliability of its annotation system.

### 5.2 BASELINE METHODS AND METRICS

Existing approaches have not systematically addressed quantitatively constrained CAD modeling. This study establishes three baseline categories: (1) Sequential methods with condition-aware LSTM architectures Sundermeyer et al. (2012); (2) Transformer-based framework Vaswani et al. (2017); Gu et al. (2017), including both its autoregressive and non-autoregressive conditional variants, follows the research direction of reconstructing CAD models from multimodal inputs (such as point clouds, sketches, text, etc.) Kolodiazhnyi et al. (2025). To ensure fair comparison, this study replaces the original multimodal inputs with quantitative constraints and converts the output from various CAD representation formats into the parametric CAD sequence splicing form proposed in this paper; (3) Specialized generative models including geometry-constrained DeepCAD and guided-diffusion D3PM Austin et al. (2021); Wu et al. (2021), forming a comprehensive evaluation system for parametric CAD generation under quantitative constraints.

The evaluation uses three core metrics—mean squared error (MSE), mean absolute error (MAE), and Pearson correlation coefficient (PCC)—computed based on reference geometric properties (e.g., surface area and volume) and the corresponding values derived from the generated CAD models. MSE captures critical deviations for manufacturing, MAE shows the average error, and PCC measures linear correlation with targets. By testing on both single- and multi-condition tasks, we systematically evaluate model performance under different constraint complexities.

### 5.3 RESULT

**Performance under single-condition constraints.** We first evaluate the generation performance under single geometric constraints (surface area or volume). As shown in Table 1, our method consistently outperforms all baseline models across all metrics. In the surface area constraint task, com-

Table 1: Comparison of CAD generation under single-property supervision. Models are evaluated independently under surface area and volume constraints.

| (a) Surface area constraints | | | | (b) Volume Constraints | | | |
|---|---|---|---|---|---|---|---|
| **Method** | MSE ↓ | MAE ↓ | PCC ↑ | **Method** | MSE ↓ | MAE ↓ | PCC ↑ |
| LSTM | 20.623 | 3.3107 | 0.7209 | LSTM | 0.2470 | 0.4134 | 0.7513 |
| Transformer(Non-AR) | 30.807 | 2.1219 | 0.5582 | Transformer(Non-AR) | 0.4160 | 0.4308 | 0.6085 |
| DeepCAD | 8.8662 | 1.8997 | 0.6120 | DeepCAD | 0.4301 | 0.3707 | 0.4891 |
| D3PM | 45.194 | 0.9814 | 0.6790 | D3PM | 1.5751 | 0.0861 | 0.7703 |
| Transformer(AR) | 1.3033 | 0.7678 | 0.8410 | Transformer(AR) | 0.0685 | 0.1097 | 0.6645 |
| Ours | **1.2763** | **0.6977** | **0.8652** | Ours | **0.0380** | **0.0631** | **0.8411** |

Table 2: Comprehensive evaluation of CAD generation under multi-condition supervision, where both surface area and volume are jointly used as constraints.

| | Surface Area Constraints | | | Volume Constraints | | |
|---|---|---|---|---|---|---|
| Method | MSE ↓ | MAE ↓ | PCC ↑ | MSE ↓ | MAE ↓ | PCC ↑ |
| LSTM | 16.328 | 3.2370 | 0.6101 | 1.0003 | 0.7189 | 0.3977 |
| Transformer(Non-Autoregressive) | 11.083 | 2.3903 | 0.5592 | 0.4057 | 0.4253 | 0.5254 |
| DeepCAD | 12.708 | 2.4851 | 0.6091 | 0.6371 | 0.4642 | 0.5477 |
| D3PM | 8.0044 | 0.5789 | 0.7762 | 0.2080 | 0.0655 | 0.8062 |
| Transformer(Autoregressive) | 1.3154 | 0.4098 | 0.8338 | 0.0621 | 0.0629 | 0.6399 |
| Ours | **0.5331** | **0.3913** | **0.9377** | **0.0072** | **0.0337** | **0.9651** |

Table 3: Ablation study on annealing probabilities (NA: non-annealed full-guidance mode).

| Method (Training) | Surface Area | | | Volume | | | Time (h) |
|---|---|---|---|---|---|---|---|
| | MSE | MAE | PCC | MSE | MAE | PCC | |
| KGBFN(NA) | 0.1596 | 0.1597 | 0.9820 | 0.0056 | 0.0202 | 0.9790 | 370 |
| KGBFN($\lambda = 0.55$) | 0.1600 | 0.1875 | 0.9805 | 0.0048 | 0.0231 | 0.9795 | 251 |
| KGBFN($\lambda = 0.35$) | 0.1288 | 0.1709 | 0.9845 | 0.0039 | 0.0212 | 0.9819 | 237 |
| KGBFN($\lambda = 0.15$) | 0.1507 | 0.1788 | 0.9816 | 0.0043 | 0.0221 | 0.9822 | 199 |

pared to previous best method (underlined in Table 1), our model reduces MSE by 2.1%, decreases MAE by 9.1%, and improves PCC by 2.9%. In the volume constraint task, the improvements are even more pronounced, with MSE reduced by 44.5%, MAE decreased by 26.7%, and PCC increased by 9%. These results clearly demonstrate the effectiveness of our model in accurately mapping CAD parametric sequences to geometric constraints, significantly enhancing generation quality.

In parametric CAD sequence generation, conventional baseline models fail to achieve dynamic constraint enforcement during generation and lack the capability to discern optimal refinement directions. Experimental results demonstrate that DeepCAD and LSTM models exhibit insufficient generation stability, while D3PM and non-autoregressive Transformers show room for improvement in geometric accuracy. In contrast, our knowledge-guided method solves these issues, reliably producing precise CAD models that meet geometric constraints with minimal supervision.

**Performance under multi-condition constraints.** We further evaluates the model performance under multi-constraint conditions (both surface area and volume). As shown in Table 2, our method demonstrates significant advantages over existing approaches: achieving over 60% improvement in MSE compared to the best baseline, along with over a 10-point gain in correlation coefficient. These results confirm the model's capability to accurately establish complex mappings between parametric CAD sequences and multiple geometric constraints.

Comparative experiments show that while baseline models like D3PM capture general trends, they fall short in generation accuracy. Transformer-based methods, in particular, show significant performance degradation when subjected to multiple constraints. These findings validate our framework's unique advantages in handling multivariate constraints, providing a reliable solution for practical multiobjective CAD design scenarios.

## 5.4 ABLATION STUDY

**Impact of the annealing routing mechanism.** We conduct ablation studies under multi-constraint settings (Table 3) to evaluate the annealing routing mechanism, focusing on the impact of reducing

Table 4: Ablation study on knowledge guidance.

| Method (Inference) | Surface Area | | | Volume | | | Time (s) |
|---|---|---|---|---|---|---|---|
| | MSE↓ | MAE↓ | PCC↑ | MSE↓ | MAE↓ | PCC↑ | |
| BFN | 1.1745 | 0.4935 | 0.8956 | 0.0340 | 0.0577 | 0.8986 | 1390 |
| KGBFN | 0.5331 | 0.3913 | 0.9377 | 0.0072 | 0.0337 | 0.9651 | 2266 |

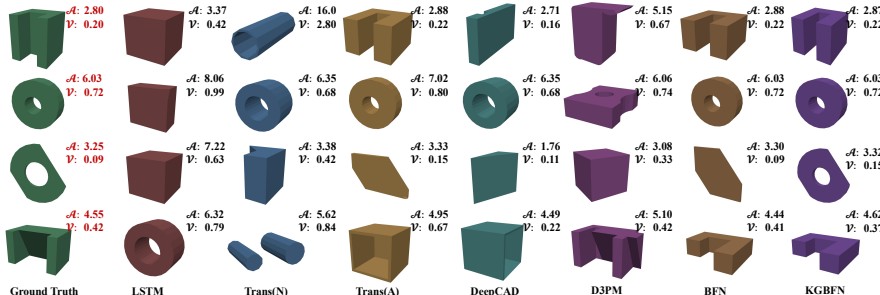

Figure 4: CAD sequence generation results under dual constraints (surface area and volume). Each row corresponds to a distinct target parameter combination.

knowledge guidance frequency during training (controlled by the annealing probability bound $\lambda$, where NA means full guidance without annealing). Using a subset with limited maximum CAD sequence length, we systematically evaluate guidance reduction. Starting from the NA (no annealing) mode, we progressively decrease guidance intensity to $\lambda$ values ranging from 0.55 to 0.15. The model maintains minimal performance fluctuation while significantly reducing training time.

**Impact of the knowledge-guided mechanism.** This study conducts ablation experiments under multi-constraint conditions to evaluate the contribution of knowledge guidance strategies (Table 4). Comparisons between the baseline without knowledge guidance (BFN) and our auxiliary knowledge-guided inference approach (KGBFN) reveal that knowledge guidance enhances the model's ability to map conditions to generation targets. During inference, the auxiliary knowledge-guided strategy achieves optimal performance. Specifically: 1) It reduces MSE and MAE for surface area and volume by up to 40% compared to the baseline; 2) Pearson correlation coefficients improve by 0.04 for surface area and 0.06 for volume.

## 5.5 CASE STUDY

This study compares four methods using target parameters $(\mathcal{A}, \mathcal{V})$ to evaluate how well they satisfy constraints and produce plausible geometries (Figure 4). The first column shows ground-truth CAD models as a reference. The other columns show results from different methods under the same constraints, along with their actual calculated parameters.

The analysis reveals clear limitations: LSTM, DeepCAD, and D3PM have significant geometric errors; the non-autoregressive Transformer yields unstable outputs; and while the autoregressive Transformer and BFN perform better, their structures remain implausible. In contrast, our method (KGBFN) achieves high accuracy and structurally valid models, with deviations under 5% from the target constraints, confirming its effectiveness for constraint-aware generation.

## 6 CONCLUSION

This paper proposes Knowledge-Guided Bayesian Flow Networks (KGBFN), a generative framework for high-fidelity parametric CAD sequences under quantitative constraints. Unlike prior methods, we use a natural language-inspired representation that concatenates parameterized instructions into continuous token sequences, enhancing dependency modeling. KGBFN integrates constraints via knowledge-guided Bayesian updates, employing a dual-channel structure with annealing to balance performance and efficiency. Experiments on CAD data with area/volume constraints show KGBFN outperforms existing methods in both single- and multi-condition tasks. This demonstrates the value of knowledge guidance for satisfying quantitative constraints.

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

## A  PROOF OF THEOREM 1

Let $C(\cdot)$ be a function that measures the desired property of the data and $V(\cdot)$ be a known function. If $Corr(C(\cdot), V(\cdot)) \geq 0$, we have:

$$D_{KL}\big(p_S(\boldsymbol{y}|x, \alpha, C) \,\|\, p_R(\boldsymbol{y}|\boldsymbol{\theta}, \alpha, C, V)\big)$$
$$\leq D_{KL}\big(p_S(\boldsymbol{y}|x, \alpha, C) \,\|\, p_R(\boldsymbol{y}|\boldsymbol{\theta}, \alpha, C)\big) \tag{12}$$

*Proof.* We begin by defining two model families:

$$\mathcal{Q}_1 := \{p(\boldsymbol{y} \mid \boldsymbol{\theta}, \alpha, C)\}, \quad \mathcal{Q}_2 := \{p(\boldsymbol{y} \mid \boldsymbol{\theta}, \alpha, C, V)\}$$

Clearly, $\mathcal{Q}_1 \subseteq \mathcal{Q}_2$, since any distribution conditioned on $C$ can be represented as a special case of the family conditioned on both $C$ and $V$ (where $V$ is effectively ignored).

By the projection property of Kullback-Leibler (KL) divergence, for any target distribution $P$ and nested model families $\mathcal{Q}_1 \subseteq \mathcal{Q}_2$, we have:

$$\min_{Q \in \mathcal{Q}_2} D_{\mathrm{KL}}(P \parallel Q) \leq \min_{Q \in \mathcal{Q}_1} D_{\mathrm{KL}}(P \parallel Q)$$

Applying this to our case with $P = p_S(\boldsymbol{y} \mid x, \alpha, C)$, we obtain:

$$D_{\mathrm{KL}}\big(p_S(\boldsymbol{y} \mid x, \alpha, C) \parallel p_R(\boldsymbol{y} \mid \boldsymbol{\theta}, \alpha, C, V)\big)$$
$$\leq D_{\mathrm{KL}}\big(p_S(\boldsymbol{y} \mid x, \alpha, C) \parallel p_R(\boldsymbol{y} \mid \boldsymbol{\theta}, \alpha, C)\big) \tag{13}$$

Here, there is a correlation between $C$ and $V$. This assumption ensures that the additional variable $V$ provides relevant information about $C$, thereby enhancing the model's capacity to approximate the true distribution. In other words, the extended model family $\mathcal{Q}_2$ has strictly greater or equal expressive power compared to $\mathcal{Q}_1$, with equality in KL divergence holding only when $V$ provides no additional information (i.e., $C \perp V \mid \boldsymbol{y}, \alpha$). $\qquad\square$

## B  TRADITIONAL BFN ALGORITHMS

This section presents the complete pseudocode for both training and inference procedures in the traditional Bayesian Flow Networks (BFN) framework.

### B.1  TRADITIONAL BFN TRAINING PROCEDURE

Algorithm 1 describes the training process for traditional BFNs with discrete data. The key steps involve:

- **Timestep scheduling**: Randomly select a timestep from the training step range, typically using uniform sampling (line 4-5).
- **Data perturbation**: Inject Gaussian noise with intensity corresponding to the current timestep into the input data to create observation samples (line 6-8).
- **Network prediction**: Feed the noisy samples into the model, requiring the network to predict the original data based on the noise level (line 9-10).
- **Loss calculation**: Compute the KL divergence between the network output and the observation samples as the loss (line 11).

### B.2  TRADITIONAL BFN INFERENCE PROCEDURE

The inference process, shown in Algorithm 2, demonstrates how to generate samples from the trained BFN model. Notable aspects include:

- **Uniform probability initialization**: At the start of inference, uniform probability mass is assigned to all possible discrete states. For discrete data with K categories, each class is initialized with probability 1/K, corresponding to starting from the maximum entropy distribution (line 3).
- **Multi-step iterative refinement**: The probability distribution is progressively refined through a reverse-time process: (1) At each timestep, the probability distribution is adjusted based on the network's predicted scores. (2) Bayesian update rules are applied to combine prior distribution with likelihood estimates (line 4).

---

**Algorithm 1:** Traditional BFN training algorithm

---

1: **Require:** initial step size $\beta(1) > 0$, steps $n$, classes $K$, denoising network $\Phi$
2: **Input:** discrete data $x \in \{1, K\}^D$, Area, Vol
3: **Output:** no output
4: $i \sim U\{1, n\}$
5: $t \leftarrow \frac{i-1}{n}$
6: $\boldsymbol{\beta} \leftarrow \beta(1)t^2$
7: $\boldsymbol{y}' \sim \mathcal{N}(\boldsymbol{y}|\beta(K\boldsymbol{e}_x - \mathbf{1}), \beta K \boldsymbol{I})$
8: $\boldsymbol{\theta} \leftarrow \text{softmax}(\boldsymbol{y}')$
9: $p_O(\boldsymbol{x}'|\boldsymbol{\theta}, t) \leftarrow \Phi(\boldsymbol{\theta}, t, \text{Area}, \text{Vol})$
10: $\hat{e}(\boldsymbol{\theta}, t) \leftarrow \left( \sum_k p_o^{(1)}(k|\boldsymbol{\theta}; t)\boldsymbol{e}_k, \ldots, \sum_k p_o^{(D)}(k|\boldsymbol{\theta}; t)\boldsymbol{e}_k \right)$
11: $L(x) \leftarrow K\beta(1)t||\boldsymbol{e}_x - \hat{e}(\boldsymbol{\theta}, t)||^2$

---

**Algorithm 2:** Traditional BFN inference algorithm

---

1: **Input:** the parameter $\boldsymbol{\theta} \in [0, 1]^{D \times K}$ and the time $t \in [0, 1]$, Area, Vol
2: **Output:** output distribution $p_O(\boldsymbol{x}|\boldsymbol{\theta}, t) \in [0, 1]^{D \times K}$
3: $\boldsymbol{\theta}_0 \leftarrow (\frac{1}{K})^{D \times K}$
4: **for** $i = 0$ *to* $N - 1$ **do**
$\quad t \leftarrow \frac{i-1}{N}$;
$\quad k \sim \Phi(\boldsymbol{\theta}_i, t, \text{Area}, \text{Vol})$;
$\quad \alpha \leftarrow \beta(1)\frac{2t'+\frac{1}{n}}{n}$;
$\quad \boldsymbol{y} \sim \mathcal{N}(\boldsymbol{y}|\alpha(K\boldsymbol{e}_k - \mathbf{1}), \alpha K \boldsymbol{I})$ ;
$\quad \boldsymbol{\theta}_{i+1} \leftarrow \frac{\boldsymbol{e}^y \boldsymbol{\theta}_i}{\sum_k (\boldsymbol{e}^y \boldsymbol{\theta}_i)_k}$;
5: $\boldsymbol{y} \in \{1, K\}^D \sim \Phi(\boldsymbol{\theta}_N, t, \text{Area}, \text{Vol})$
6: **Return** $\boldsymbol{y}$

---

- **Final distribution sampling**: After completing all timestep iterations: (1) Samples are drawn from the final optimized probability distribution. (2) For deterministic outputs, the category with maximum probability can be selected directly. (3) The sampling process strictly adheres to the data distribution characteristics learned by the model (line 5).

## C  KNOWLEDGE-GUIDED BFN ALGORITHMS

This section presents the complete pseudocode for our proposed Knowledge-Guided Bayesian Flow Networks (KGBFN). This algorithm proposes three core technical innovations: (1) The knowledge guidance mechanism computes domain knowledge through $V(\cdot)$, providing real-time constraint information for both inference and training processes; (2) The dual-channel architecture maintains a standard BFN primary pathway and a knowledge-enhanced secondary channel to achieve knowledge fusion; (3) The annealed activation judgment dynamically controls knowledge injection timing through annealed probability sampling.

### C.1  KGBFN TRAINING PROCEDURE

Algorithm 3 describes the enhanced training process with knowledge guidance. Key improvements over traditional BFN include:

- **Timestep sampling**: (1) Sample timestep t from a uniform distribution (line 4-5). (2) Determine the current noise scheduling parameters and the probability of knowledge integration (line 6, 9).

- **Primary pathway computation**: (1) Perform forward BFN process to generate noisy observations (line 7-8). (2) Predict original data distribution conditioned on noise levels (line 10-11). (3) Compute KL divergence loss between predicted and true distributions (line 12).

---

**Algorithm 3:** KGBFN training algorithm

---

1: **Require:** initial step size $\beta(1) > 0$, steps $n$, classes $K$, denoising network $\Phi$, and binary sampler Prob_Sample($\cdot$)

2: **Input:** discrete data $x \in \{1, K\}^D$, Area, Vol

3: **Output:** no output

4: $i \sim U\{1, n\}$

5: $t \leftarrow \frac{i-1}{n}$

6: $\boldsymbol{\beta} \leftarrow \beta(1)t^2$

7: $\boldsymbol{y}' \sim \mathcal{N}(\boldsymbol{y}|\boldsymbol{\beta}(Ke_x - \mathbf{1}), \beta K\boldsymbol{I})$

8: $\boldsymbol{\theta} \leftarrow \text{softmax}(\boldsymbol{y}')$

9: **Judge** $\leftarrow 0$

10: $p_O(x'|\boldsymbol{\theta}, t) \leftarrow \Phi(\boldsymbol{\theta}, t, \text{Area}, \text{Vol}, \textbf{Judge}, 0)$

11: $\hat{e}(\boldsymbol{\theta}, t) \leftarrow \left( \sum_k p_o^{(1)}(k|\boldsymbol{\theta};t)\boldsymbol{e}_k, \ldots, \sum_k p_o^{(D)}(k|\boldsymbol{\theta};t)\boldsymbol{e}_k \right)$

12: $L_1(x) \leftarrow K\beta(1)t||\boldsymbol{e}_x - \hat{e}(\boldsymbol{\theta}, t)||^2$

13: **Judge** $\leftarrow$ Prob_Sample($t$)

14: $\alpha \leftarrow \beta(1)\frac{2t + \frac{1}{n}}{n}$

15: $k' \sim p_O(\boldsymbol{x}'|\boldsymbol{\theta}, t)$

16: $\boldsymbol{y} \sim \mathcal{N}(\boldsymbol{y}|\alpha(Ke_{k'} - \mathbf{1}), \alpha K\boldsymbol{I})$

17: $\boldsymbol{\theta}_{i+1} \leftarrow \frac{e^y \boldsymbol{\theta}_i}{\sum_k (e^y \boldsymbol{\theta}_i)_k}$

18: $t' \leftarrow t + \frac{1}{n}$

19: $p_O(x'|\boldsymbol{\theta}_{i+1}, t') \leftarrow \Phi\left( \boldsymbol{\theta}_{i+1}, t', \text{Area}, \text{Vol}, \textbf{Judge}, V(k, \text{Area}, \text{Vol}) * \textbf{Judge} \right)$

20: $\hat{e}(\boldsymbol{\theta}_{i+1}, t') \leftarrow \left( \sum_k p_o^{(1)}(k|\boldsymbol{\theta}_{i+1};t')\boldsymbol{e}_k, \ldots, \sum_k p_o^{(D)}(k|\boldsymbol{\theta}_{i+1};t')\boldsymbol{e}_k \right)$

21: $L_2(x) \leftarrow K\beta(1)t||\boldsymbol{e}_x - \hat{e}(\boldsymbol{\theta}_{i+1}, t')||^2$

22: $L(x) \leftarrow L_1(x) + \lambda * L_2(x)$

---

- **Knowledge guidance mechanism**: (1) Control knowledge injection timing via annealing probability (line 13). (2) Encode primary pathway outputs to compute knowledge representation V (line 14-19). (3) Fuse observations with V through neural network to compute knowledge-enhanced loss (line 19-20).

- **Joint training**: Optimize network parameters end-to-end via backpropagation using weighted combined losses from both pathways (line 21-22).

## C.2 KGBFN INFERENCE PROCEDURE

Algorithm 4 details the knowledge-enhanced inference process. Notable features include:

- **Uniform probability initialization**: At the start of inference, uniform probability mass is assigned to all possible discrete states. For discrete data with K categories, each class is initialized with probability 1/K, corresponding to starting from the maximum entropy distribution. (line 3)

- **Multi-step iterative refinement**: The probability distribution is progressively refined through a reverse-time process: (1) Use annealing probability to determine whether to use the main channel or auxiliary channel for denoising at each step. (2) If using the auxiliary channel, compute the corresponding knowledge. (3) Adjust the probability distribution using the denoising network (line 4). (4) Apply Bayesian update rules to combine prior distribution with likelihood estimates (line 10).

- **Final distribution sampling**: After completing all timestep iterations: (1) Draw samples from the final optimized probability distribution. (2) For deterministic outputs, the category with maximum probability can be selected directly. (3) The sampling process strictly adheres to the data distribution characteristics learned by the model (line 11).

---

**Algorithm 4:** KGBFN inference algorithm

---

1: **Input:** the parameter $\boldsymbol{\theta} \in [0,1]^{D \times K}$ and the time $t \in [0,1]$, Area, Vol
2: **Output:** output distribution $p_O(\boldsymbol{x}|\boldsymbol{\theta},t) \in [0,1]^{D \times K}$
3: $\boldsymbol{\theta}_0 \leftarrow (\frac{1}{K})^{D*K}$
4: $t \leftarrow 0$
5: **Judge** $\leftarrow 0$
6: $k \sim \Phi(\boldsymbol{\theta}_0, t, \text{Area}, \text{Vol}, \textbf{Judge}, 0)$
7: $\alpha \leftarrow \beta(1)\frac{2t'+\frac{1}{n}}{n}$
8: $\boldsymbol{y} \sim \mathcal{N}(\boldsymbol{y}|\alpha(K\boldsymbol{e}_k - \mathbf{1}), \alpha K\boldsymbol{I})$
9: $\boldsymbol{\theta}' \leftarrow e^y\boldsymbol{\theta}$
10: $\boldsymbol{\theta}_{i+1} \leftarrow \frac{\boldsymbol{\theta}'}{\sum \boldsymbol{\theta}'_k}$ **for** $i = 1$ *to N-1* **do**
   | $\quad$ **Judge** $\leftarrow$ Prob_Sample($t$) ;
   | $\quad t \leftarrow \frac{i-1}{N}$ ;
   | $\quad k \sim \Phi(\boldsymbol{\theta}_i, t, \text{Area}, \text{Vol}, \textbf{Judge}, V(k, \text{Area}, \text{Vol}) * \textbf{Judge})$ ;
   | $\quad \alpha \leftarrow \beta(1)\frac{2t'+\frac{1}{n}}{n}$ ;
   | $\quad \boldsymbol{y} \sim \mathcal{N}(\boldsymbol{y}|\alpha(K\boldsymbol{e}_k - \mathbf{1}), \alpha K\boldsymbol{I})$ ;
   | $\quad \boldsymbol{\theta}_{i+1} \leftarrow \frac{e^y\boldsymbol{\theta}_i}{\sum_k (e^y\boldsymbol{\theta}_i)_k}$ ;
11: $\boldsymbol{y} \in \{1, K\}^D \sim \Phi(\boldsymbol{\theta}_N, t, \text{Area}, \text{Vol}, \textbf{Judge}, V(k, \text{Area}, \text{Vol}) * \textbf{Judge})$
12: **Return** $\boldsymbol{y}$

---

# D IMPLEMENTATION DETAILS

## D.1 TRAINING CONFIGURATION

The traditional BFN denoising network was trained on a single RTX 3090 GPU using the Adam optimizer with a learning rate of $1 \times 10^{-6}$ and a batch size of 128 for approximately 12 days. An exponential moving average (EMA) of the model parameters was maintained during training with a decay rate of 0.0001. The knowledge-guided BFN was trained under the same settings, using the Adam optimizer, the same learning rate and batch size, and EMA with a decay of 0.0001, for about 18 days on a single RTX 3090 GPU.

## D.2 NETWORK ARCHITECTURE

We employ a decoder-based Transformer model as the denoising network $\Phi$ to model the probability distribution of CAD token sequences. This network architecture consists of 21 stacked Transformer layers, with each layer incorporating 16 parallel attention mechanisms and a 1024-dimensional feature representation space. The input tokens are first mapped to a 1024-dimensional embedding space and fused with adaptively learned positional encoding information.

The model adopts the classic Transformer implementation, integrating layer normalization operations and residual skip connections. Unlike sequential autoregressive models, our approach removes the attention mask constraints, enabling global attention connections among all tokens in the sequence. At the output end of the network, the model constructs a categorical probability distribution over a predefined set of 263 discrete tokens. This design facilitates collaborative optimization of the entire sequence during the denoising generation process.

## D.3 ACTUAL SEQUENCE LENGTH DISTRIBUTION

To gain deeper insights into the structural characteristics of CAD sequences in our dataset, we conducted an analysis of the actual sequence length distribution (excluding padding tokens) in the training set. As illustrated in Figure 5, the sequence lengths exhibit a distribution ranging from 16 to 64 tokens, comprehensively covering CAD models of varying complexity levels. This distribution pattern demonstrates that our dataset encompasses a complete spectrum of CAD modeling cases, from basic short sequences to sophisticated long sequences.

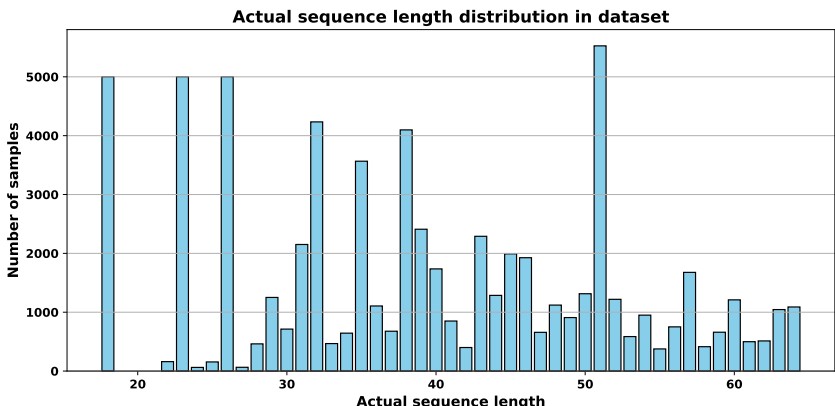

Figure 5: Actual sequence length distribution in the training set after removing padding tokens. The x-axis shows the true sequence lengths, and the y-axis shows the number of samples.

### D.4 IMPLEMENTATION DETAILS OF BASELINE MODEL

The dataset used for model training is constructed from preprocessed parametric CAD files following deepCAD. Each CAD model is parsed into a sequence of operation commands — including L (Line), A (Arc), R (Circle), and E (Extrusion) — with each command followed by its associated parameters. These command-parameter pairs are sequentially concatenated to construct the final CAD token sequence. This sequence modeling method preserves the detailed information of CAD operations to the greatest extent, enabling the model to better capture the semantic and structural patterns within the sequence. This dataset construction method is adopted by several baseline models, including Transformer (autoregressive), Transformer (non-autoregressive), BFN, D3PM, and LSTM.

Unlike our proposed method, which converts each CAD model into a flat sequence of minimal tokens by decomposing all instructions into their smallest components, the DeepCAD baseline treats each instruction as a single token. These instruction-level tokens, which are inherently of variable length due to differing argument structures, are embedded and padded to form fixed-length sequences for input to the model. This approach results in a coarser-grained representation compared to our method, which operates at a finer granularity and retains more detailed operational semantics.

**Transformer (AR)** The sequence-to-sequence Transformer autoregressive architecture used in our baseline model is specifically designed for parametric CAD sequence generation tasks. The model consists of a 4-layer Transformer encoder and a 4-layer Transformer decoder, with a hidden dimension of 256. The encoder takes as input both the normalized surface area and volume—each projected through a linear layer and replicated across the sequence dimension—as well as the embeddings of original CAD command sequences. This allows the encoder to integrate geometric constraints with the raw sequence information to build rich contextual representations. The decoder performs autoregressive prediction based on the encoder output. It uses a masking mechanism to ensure each decoding step only attends to previously generated tokens. The vocabulary size is 263, including special tokens for padding, beginning-of-sequence (BOS), and end-of-sequence (EOS). All input sequences are either truncated or padded to a fixed maximum length of 64.

**Transformer (NA)** The baseline model adopts a non-autoregressive autoencoder (AE) Transformer architecture designed for efficient reconstruction and generation of parametric CAD sequences. The model consists of a 4-layer Transformer encoder and a 4-layer Transformer decoder, with a hidden dimension of 256, 8 attention heads, and a feedforward dimension of 512. A dropout rate of 0.1 is applied throughout the model to prevent overfitting. The model takes normalized surface area and volume as geometric constraints, which are first projected into 256-dimensional feature vectors via a linear layer and then replicated along the sequence length dimension as inputs to the encoder. This design allows the model to effectively incorporate global geometric attributes during encoding. The decoder receives CAD command sequences embedded with positional information, which are combined with the encoded geometric constraint features and then used to perform parallel predic-

tion to reconstruct the full CAD operation sequence. In this way, geometric constraints directly influence sequence generation during decoding. Compared to traditional autoregressive models, this non-autoregressive design significantly improves inference speed and is well-suited for large-scale CAD sequence generation tasks.

**LSTM** The LSTM baseline model is designed for parametric CAD sequence generation with conditional inputs representing geometric constraints. Specifically, the model conditions on two features—normalized surface area and volume—which are fed as inputs to initialize the LSTM's hidden state. The model uses a hidden size of 512 and a vocabulary size of 265, processing fixed-length sequences of length 64. Training is performed with a batch size of 1000, employing 8 worker threads for data loading and pinned memory for efficient GPU transfer. The optimizer is Adam with an initial learning rate of 1e-2, betas set to (0.9, 0.98), and weight decay of 0.01. The training schedule includes 3 million steps, with checkpoints and validation performed every 100 steps.

**D3PM** The D3PM baseline model employs a diffusion probabilistic model with absorbing states, specifically designed for discrete token sequence generation. The architecture uses a vocabulary size of 263 and token embeddings of dimension 768, with 12 Transformer blocks and 12 attention heads (calculated as 768 divided by 64). The model input consists of embedded CAD token sequences along with a 128-dimensional conditioning vector, which is derived from normalized surface area and volume through a specific mapping to incorporate geometric constraints. The token sequences are mapped into a 768-dimensional embedding space and are input to the model jointly with the conditioning vector. The diffusion process runs for 1000 timesteps (T=1000) with a cross-entropy loss weighting parameter of 0.05. Training is performed with a batch size of 512, an initial learning rate of 1e-3, and a minimum learning rate of 1e-5. Gradient accumulation is set to 4 steps, with a warm-up period of 2500 iterations and a maximum of 500,000 training iterations. Weight decay is set to 0.1, and evaluation occurs every 1000 iterations. The training uses a fixed random seed to ensure reproducibility. The model outputs discrete token sequences generated stepwise, aiming to produce high-quality parametric CAD sequences.

