# OpenReview forum: "Knowledge Guided Bayesian Flow Network for CAD Sequence Generation"
_ICLR.cc/2026/Conference — Submitted to ICLR 2026_

### Official Review · Reviewer_dNRF · 2025-10-30

**Soundness:** 2
**Presentation:** 3
**Contribution:** 1
**Rating:** 2
**Confidence:** 5

**Summary:**

This paper introduces the Knowledge-Guided Bayesian Flow Network (KGBFN) to address the generation of parametric CAD sequences under precise quantitative constraints (e.g., target surface area or volume). The framework represents CAD sequences as a flat, discrete token sequence, making them suitable for a Bayesian Flow Network (BFN). The use of direct, non-differentiable error feedback from a geometry engine is a clever mechanism.

**Strengths:**

1. This paper is well-written and easy to follow.

2. Most CAD generation research focuses on qualitative shape fidelity. This paper tackles the much more difficult and practical engineering problem, generating a model that adheres to precise quantitative specifications.

3. The core mechanism of injecting a direct error signal from a non-differentiable, external geometry engine back into the generative process is an innovative touch.

**Weaknesses:**

1. The paper's novelty is significantly tempered by latest work (TGBFN[*]) that addresses the identical problem with the same base model (BFN) and data representation. Besides, the performance of proposed KGBFN is much far from the TGBFN, which again further weakens the contribution of this manuscript.

2. The overview diagram shown in Figure 1 is extremely similar to the architecture diagram in TGBFN[*] in terms of composition style and icons used. The reviewer strongly suggests that the author modify and redraw Figure 1.

3. While the annealing mechanism is shown to reduce training time, the inference process for the final KGBFN model is still computationally expensive. Table 4 shows that KGBFN inference (2266s) is ~63% slower than the standard BFN baseline (1390s), which could be a barrier to practical, real-time applications. Meanwhile, the inference time of TGBFN[*] could run in round 12s, which significantly surpasses the proposed KGBFN.

4. The general approach of using BFNs for quantitatively-constrained CAD generation is not unique. The latest paper "Target-Guided Bayesian Flow Networks" (TGBFN[*])  tackles the identical problem (BFN for CAD under area/volume constraints) and also discretizes the data. While the method of guidance differs (direct error-signal injection in KGBFN vs. a learned posterior guidance model in TGBFN), the core problem setup and base model are the same, which significantly reduces the paper's fundamental originality.

5. The current framework is only demonstrated on global constraints (total surface area, total volume). It is unclear how this method would scale to more complex and common local constraints (e.g., "the radius of this specific hole must be 10mm"). The current global error-vector feedback signal seems insufficient for such fine-grained, local control.

[*] Wenhao Zheng, Chenwei Sun, Wenbo Zhang, Jiancheng Lv, and Xianggen Liu. Target-Guided Bayesian Flow Networks for Quantitatively Constrained CAD Generation. In Proceedings of the 33rd ACM International Conference on Multimedia (MM '25). Association for Computing Machinery, New York, NY, USA, 3330–3339. https://doi.org/10.1145/3746027.3755052

**Questions:**

1. Considering the proposed whole framework of KGBFN is very similar to TGBFN[*]. Can the authors clearly state the advantages of their "direct error injection" method compared to TGBFN's "learned guided posterior" method? Why is one approach superior to the other? From an experimental perspective, the performance of KGBFN is far inferior to that of TGBFN.

[*] Wenhao Zheng, Chenwei Sun, Wenbo Zhang, Jiancheng Lv, and Xianggen Liu. Target-Guided Bayesian Flow Networks for Quantitatively Constrained CAD Generation. In Proceedings of the 33rd ACM International Conference on Multimedia (MM '25). Association for Computing Machinery, New York, NY, USA, 3330–3339. https://doi.org/10.1145/3746027.3755052

---

### Official Review · Reviewer_1Z6b · 2025-10-30

**Soundness:** 3
**Presentation:** 3
**Contribution:** 2
**Rating:** 4
**Confidence:** 4

**Summary:**

This paper proposes the Knowledge-Guided Bayesian Flow Network (KGBFN) for generating parametric CAD sequences under quantitative constraints (e.g., surface area, volume). Its core innovation is a "knowledge-guided Bayesian update." During denoising, the model periodically uses an external CAD tool (PythonOCC) to compute the geometric properties of intermediate sequences. The resulting property deviation is then injected back as "knowledge" to guide generation. To manage the high computational cost, the authors use a dual-channel architecture with an annealing mechanism to balance guidance frequency and efficiency. Experiments on a CAD subset show KGBFN outperforms baselines on single and multi-constraint tasks.

**Strengths:**

The key contribution is integrating non-differentiable domain knowledge directly into the BFN's iterative process. By rendering and calculating property deviations mid-generation, the model gets explicit feedback on hard constraint satisfaction, a clever approach for constrained generation.
The authors pragmatically address the computational bottleneck of the PythonOCC guidance. The proposed dual-channel architecture and annealing mechanism allow the model to dynamically skip expensive guidance steps, achieving a reasonable trade-off between performance and efficiency, as validated by ablation studies.

**Weaknesses:**

All experiments use a highly simplified dataset. Real-world CAD models have far longer sequences. The examples shown are simple extrusions, not "complex geometric models." There is no evidence this method scales to complex, long-sequence tasks.

The core mechanism relies on repeatedly calling PythonOCC for B-Rep rendering. This is extremely expensive: inference time increased significantly (Table 4) and training took 200-370 hours (Table 3) for simple tasks. The method is computationally unfeasible for longer sequences, limiting its practical value.

**Questions:**

Can you provide performance data on longer, more complex CAD sequences (e.g., length > 200)? how do you assess the computational feasibility of this method for real-world tasks?
the annealing probability is not deeply analyzed.which stage of the denoising process is knowledge guidance most critical?
The experiments are limited to global properties (area, volume). What about others?

---

### Official Review · Reviewer_9Yuq · 2025-10-31

**Soundness:** 3
**Presentation:** 2
**Contribution:** 2
**Rating:** 2
**Confidence:** 5

**Summary:**

In the field of CAD generation, this paper proposes a knowledge guided Bayesian flow network, which, on the basis of the traditional Bayesian flow network, introduces the guidance of surface area and volume through an annealing mechanism, improving controllability over surface area and volume. However, this paper lacks originality, contains some ambiguous technical points in its writing, and the method has issues at the practical application level. Besides,  the experiments in the paper are insufficient. The specific comments are presented in the Strengths and Weaknesses parts.

**Strengths:**

1. The paper is well-organized and easy to follow.

2. The CAD sequence generation problem is interesting.

**Weaknesses:**

- Novelty:

(1) This paper shows a very large similarity to TGBFN [1] in both figures and methodology; both add control information of surface area and volume on top of BFN, and the originality is very limited.

- Methods:

(1) Despite the annealing mechanism, the time for PythonOCC to compute surface area and volume is still too long, leading to defects of the method in practical applications.

(2) The paper uses BFN as the basic generation method, but lacks a theoretical comparison with D3PM and a comparison of generation results with D3PM under the same experimental conditions, i.e., knowledge guiding based on the annealing mechanism.

(5) In the early stage of inference when noise is relatively large, the CAD sequence tends to be invalid, and it may not be possible to compute volume and surface area through PythonOCC. The paper does not specifically explain how this situation is handled.

- Writing:

(1) The guidance information in the paper is surface area and volume; calling it “knowledge guided” is somewhat far-fetched.

(2) The paper presents the method pipeline with only one figure, resulting in rather cluttered information in the figure and some missing information. It is recommended to split this figure into multiple figures and appropriately add details such as the network architecture.

(3) The abstract mentions “jointly model discrete and continuous variables” but the subsequent content does not further explain how continuous parameters are modeled.

- Experiments:

(1) The paper does not compare with the latest CAD sequence generation methods, e.g., TGBFN [1], Seek-CAD [2], CAD-GPT [3].

(2) The paper needs to add more generated visual results to demonstrate the diversity of the generations.


- Reference:


[1]Zheng, Wenhao, et al. "Target-Guided Bayesian Flow Networks for Quantitatively Constrained CAD Generation." Proceedings of the 33rd ACM International Conference on Multimedia. 2025.

[2]Li, Xueyang, et al. "Seek-CAD: A Self-refined Generative Modeling for 3D Parametric CAD Using Local Inference via DeepSeek." arXiv preprint arXiv:2505.17702 (2025).

[3]Wang, Siyu, et al. "Cad-gpt: Synthesising cad construction sequence with spatial reasoning-enhanced multimodal llms." Proceedings of the AAAI Conference on Artificial Intelligence. Vol. 39. No. 8. 2025.

**Questions:**

(1) The proposed method uses surface area and volume as control conditions; however, in practical applications it is rare to use surface area and volume as control conditions. Could methods that use more common controls such as bounding boxes, point clouds, and sketches be further explored?

(2) Where is the advantage of guidance based on surface area and volume compared with guidance based on rendered vision?

---

### Official Review · Reviewer_VUuG · 2025-11-03

**Soundness:** 2
**Presentation:** 2
**Contribution:** 2
**Rating:** 2
**Confidence:** 2

**Summary:**

This paper focus on generating  parametric CAD sequences under geometric constraints. It proposed a knowledge-guided BFN to jointly model discreate and continuous parameters in the CAD sequences. The rendered geometric property feedback is computed from intermediate steps.

**Strengths:**

Authors proposed knowledge-guided and dual channel BFN, which leads to better results. But I should say I am not an expert in BFN. And it is unclear to me why we even need BFN for this task. There is clearly a lot of diffusion and AR or combined model that can handle both continuous and discrete data. I don't see a clear advantage for BFN in this case.

**Weaknesses:**

Task formulation is a bit weird, usually when it comes to CAD constraints, one think of geometry or topology constraints. E.g two holes should be equal radius, the two planes should be parallel to each other, the holes should be positioned here for connectivity etc. Directly using volume and surface area to control overall generation is definitely a less common task. It is also more interesting to understand how the constraint enable reasoning or CoT during generation, as oppossed to directly using it as a condition as in the paper.

**Questions:**

Since I am not an expert on Bayesian Flow Networks, I will leave detailed questions about the network design and implementation to other reviewers. From a CAD generation perspective, the visual results are quite limited . Examples shown are mostly simple primitives such as cubes and cylinders. This level of visual complexity is insufficient to demonstrate the model’s capability for generating realistic or diverse CAD, which honestly is a more fundemantal problem that needs to be solved before controllable generation.

---

### Meta-Review · Area_Chair_D4m2 · 2026-01-10

**Summary:**

The reviewers raised consistent and substantial concerns regarding the paper’s novelty, practical relevance, experimental scope, and computational feasibility. They broadly agreed that the work is incremental relative to very recent prior art (notably TGBFN) and demonstrated only on overly simplified CAD datasets.

**Reviewer Concerns:**

Concerns addressed (partially):
- Clarified the role of the dual-channel architecture and annealing mechanism in reducing the frequency of expensive PythonOCC calls.
- Explained how non-differentiable geometric error signals are injected during denoising, addressing some methodological confusion raised by non-expert reviewers.

Concerns remain:
- Novelty relative to TGBFN.
- Experimental insufficiency and realism & practical feasibility. The use of surface area and volume as primary controls continues to be seen as atypical and weakly motivated for practical CAD workflows.
- Missing or insufficient comparisons with recent CAD generation baselines (e.g., Seek-CAD, CAD-GPT) remain unaddressed.

**Reviewer Scores:**

No reviewer appears likely to revise their score upward significantly enough.

---

### Decision · Program_Chairs · 2026-01-26

Reject